# Centrality Learning: Auralization and Route Fitting [note 1]

**DOI:** 10.3390/e25081115

**Published:** 2023-07-26

**Authors:** Xin Li, Liav Bachar, Rami Puzis

**Affiliations:** 1Department of Mechatronics Engineering, Ben-Gurion University of the Negev, Beer-Sheva 84105, Israel; lx@post.bgu.ac.il; 2Department of Software and Information Systems Engineering, Ben-Gurion University of the Negev, Beer-Sheva 84105, Israel; liavba@post.bgu.ac.il

**Keywords:** centrality, deep learning, sound recognition, routing, auralization

## Abstract

Developing a tailor-made centrality measure for a given task requires domain- and network-analysis expertise, as well as time and effort. Thus, automatically learning arbitrary centrality measures for providing ground-truth node scores is an important research direction. We propose a generic deep-learning architecture for centrality learning which relies on two insights: 1. Arbitrary centrality measures can be computed using Routing Betweenness Centrality (RBC); 2. As suggested by spectral graph theory, the sound emitted by nodes within the resonating chamber formed by a graph represents both the structure of the graph and the location of the nodes. Based on these insights and our new differentiable implementation of Routing Betweenness Centrality (RBC), we learn routing policies that approximate arbitrary centrality measures on various network topologies. Results show that the proposed architecture can learn multiple types of centrality indices more accurately than the state of the art.

## 1. Introduction

Representation learning in graphs, also known as graph embedding, has proven to be a valuable technique for enabling diverse downstream analysis tasks, including network classification [1], community detection [2], link prediction [3], and node classification [4]. However, a particularly challenging application of representation learning lies in inferring the centrality of nodes in a graph based on their positional significance within the graph topology. Centrality computation is one of the most important and commonly used tools in the analysis of complex networks [5,6]. Centrality measures quantify nodes’ importance for different tasks. For example, nodes with high betweenness centrality can be used to monitor network flows [7,8]; nodes with high closeness can easily reach all other network participants, meaning they can be used to disseminate information [9]; etc. The most prominent centrality measures are the connectivity degree (or just degree), closeness, betweenness, and eigenvector centrality.

Researchers continue to invent new centrality measures and node properties to fit a particular task which has not been properly covered yet [10,11]. Formulating new structural node properties, including centrality measures, for a task at hand requires domain expertise, graph-analysis expertise, and, of course, time and effort. It is important to devise a generic method capable of learning centrality measures for providing the ground-truth importance of nodes.

Conventional machine learning techniques have been employed to estimate centrality measures by utilizing pre-existing centrality measures as features [12,13,14]. However, this approach inherits the strengths and limitations of the centrality measures used as input features and may not be capable of learning a novel notion of centrality. In order to address this limitation, efforts have been made to leverage Graph Neural Networks (GNNs) for learning centrality. Maurya et al. [15] proposed GNN architectures that do not rely on pre-computed centrality measures. Unfortunately, closeness and betweenness measures are learned using tailor-fit GNN architectures, thus relying on human expertise. We explored a fully automated Learned Routing Centrality (LRC) method which utilizes the generic nature of routing betweenness centrality to learn arbitrary centrality measures. We demonstrated LRC on geometric graphs where routes were inferred from node positions on a 2D plane. In this article, we elaborate LRC based on spectral graph theory and the auralization of nodes.

Spectral graph analysis is widely recognized as a valuable tool for tasks such as graph mining, comparison, and classification [16,17,18,19]. In this framework, eigenvalues of a graph correspond to the stationary frequencies, or spectrum, of heat or wave propagation within the graph. The corresponding eigenvector components are frequently employed as node representations for tasks such as node classification and clustering. Typically, this representation encompasses a subset of the stationary state associated with the top *k* eigenvalues. However, the dynamic, i.e., waveform, representation of the nodes remains underexplored. In this article, we utilize the waveform-based representation of nodes rather than their spectrum and use a sound recognition neural network to learn routes corresponding to various centrality measures.

This article is based on two approaches presented at the Complex Networks conference and a combination thereof. The following is a summary of contributions:We propose a graph auralization scheme as power iterations with momentum over the right stochastic matrix derived from the graph adjacency. The convergence of the power iterations is slowed down using momentum to provide audible waveforms.This article demonstrates a surprising combination of graph analysis with sound recognition neural networks. We show that waveform graph analysis facilitates non-trivial downstream tasks such as learning centrality from graph auralization (LCGA) [20] and hope to inspire additional architectural solutions combining graph analysis and sound analysis.We present a method to learn routing centrality (LRC) [21]. This method utilizes the generic nature of RBC to learn arbitrary centrality measures in geometric graphs by fitting a routing policy. This article also presents a novel approach for computing eigenvector-based RBC in a differentiable manner, enabling the use of gradient back-propagation in computing graphs and, thus, can be used in conjunction with deep-learning algorithms.Finally, we combine LRC and LCGA into an algorithm which learns a routing centrality based on graph auralization (LRCGA), producing the most accurate results.

The rest of the article is structured as follows: Section 2 discusses the literature related to centrality learning and provides background on RBC and sound recognition. Section 3 includes a high-level overview of the proposed algorithms and elaborates on each one of them. In Section 4, we demonstrate the algorithms’ performance and discuss the results. The article concludes with Section 5.

## 2. Background and Related Work

### 2.1. Centrality Inference Using Traditional Machine Learning

In recent years, with the advancements in the machine- and deep-learning fields, researchers have utilized many of these algorithms to approximate the centrality of graph nodes. Most works in this category infer some centrality measures from other centrality measures. In 2018, Grando et al. [12] relied on the degree and on the eigenvector centrality to learn other centrality measures. In general, they constructed a vector containing two features, where the first feature is the node’s degree centrality, whereas the latter is the node’s eigenvector centrality. They trained an NN model based on synthetic graphs and the vectors obtained for predicting the node’s centrality. Later, Grando et al. evaluated the performance of their model and compared it to other state-of-the-art ML algorithms, and applied the regression model obtained for real-world graphs.

Mendonca et al. [13] improved the work presented by Grando et al. by proposing the NCA-GE model. The NCA-GE architecture utilizes Structure2Vec [22] and Graph Convolution Network (GCN) [23] for generating a high-dimensional feature vector for each node in a given graph. To these generated node embeddings, they added the degree centrality as an additional dimension. Finally, they utilized the embeddings obtained to approximate node centrality. They evaluated their approximations on synthetic and real-world networks by measuring the mean Kendall coefficient between their approximations and the actual centrality score. They presented better correlations in comparison to Grando et al.’s work.

Zhao et al. [14] conducted a study on identifying influential nodes by considering multiple features, including nine different centrality measures. The focus of their investigation was to assess the influence of nodes, which was simulated through epidemic propagation. Utilizing pre-computed centrality measures can facilitate the inference of related measures; however, it may not effectively generalize to approximate uncorrelated measures or tasks that necessitate the learning of novel measures. Unlike traditional methods, which evaluate a node’s importance based on one or some global or local topologies, Zhao et al. proposed a framework which detects the importance of a node based on ML. The framework generated a feature vector of each node consisting of the values of nine famous and classical centralities, such as degree, closeness, eigenvector, PageRank, etc., and the infection rate (which is an essential factor in the propagation scenarios). Then, they labelled each node based on the actual propagation ability obtained from their simulated propagation, which was based on the SIR model [24]. They tested the model on different classic information scenarios and different sizes and types of networks. The results showed that, in general, ML methods outperformed the traditional centrality methods. Still, the accuracy of the ML methods was affected by both the infection rate and the number of labels. As opposed to the works of Grando et al. [12] and Mendoncca et al.  [13], this model considers the graph structure and not only fixed centrality measures.

It is important to mention that Grando et al. [12], Mendonca et al. [13], and  [14] considered other centrality measures in the pre-processing step. Thus, the models might have good results when evaluating correlated measures, but might not generalize well to approximate un-correlated measures or tasks that require learning new measures.

### 2.2. Centrality Inference Using Graph Neural Networks

Maurya et al. [15] proposed GNN-Bet and GNN-Close, two graph neural networks to approximate betweenness and closeness centralities, respectively. In GNN, a vector representation is generated to each node based on the aggregation of the adjacent nodes; therefore, these features utilized these characteristics to model paths and learn how many nodes are reachable by a specific node. The model receives the input graph and the adjacency matrix. Next, the graphs are pre-processed, and the adjacency matrix is modified such that nodes aggregate features over multiple hops along possible shortest paths in the given graph. The model learns a score function which maps the aggregated node’s information to a score correlated with the centrality measure score. Two variants of the model, GNN-Bet and GNN-Close, for approximating betweenness and closeness were proposed. GNN-Bet employs two modified adjacency matrices (incoming and outgoing) and GNN-Close only one. As a consequence, they required a different pre-processing step to prepare the matrices and different GNN structures to process them. The experiments were conducted on a series of synthetic and real-world graphs, where both models were faster than other methods while providing higher ranking performance. In addition, the model training is inductive; it can be trained on one set of graphs and evaluated on another set of graphs with varying structures.

Fan et al. [25] proposed a graph neural network encoder–decoder ranking model to identify nodes with the highest betweenness. To identify the nodes with the highest SPBC, they used a graph neural network encoder–decoder ranking model. The model first encodes the nodes into embedding vectors capturing structural information for each node in the embedding space. Then, they use a decoder designed as a multi-layer perceptron to compute the SPBC ranking score. Extensive experiments were conducted on synthetic graphs and real-world graphs from different domains. Fan et al. achieved accuracy on both synthetic and real-world graphs comparable to state-of-the-art sampling-based baselines, but their model is far more efficient than sampling-based baselines for the running time.

### 2.3. Background on Routing Betweenness Centrality

In 2010, Dolev et al. [26] proposed the Routing Betweenness Centrality (RBC) measure, which generalizes previously well-known betweenness centrality measures, such as Shortest Path Betweenness Centrality (SPBC) [27], Flow Betweenness Centrality (FBC) [28], and Load Centrality (LC) [29]. These measures assume different routing strategies. In SPBC, one shortest path between source and target is chosen uniformly at random. In LC, the routing decision is made at every hop resulting in a non-uniform probability of choosing a shortest path. FBC equally considers the paths of all lengths between source and target, like in the max-flow problem.

RBC is a flexible and realistic measure which accommodates a broad class of routing strategies. It measures the extent to which nodes or groups of nodes are exposed to the traffic given any loop-free routing strategy. Let G=(V,E) be a simple unweighted undirected graph with |V|=n nodes and |E|=m edges. Dolev et al. defined a routing policy $R:V^{4}\to R$ as the probability p=R(s,t,u,v) that the node *u* will forward to *v* the packet sent from *s* to *t*. In this article, we define the routing policy *R* as a function which computes the probability *p*, providing a representation of the nodes s,t,u,v. Possible representations will include node positions in Euclidean space (Section 3.4) or node waveforms (Section 3.5).

A traffic matrix T(s,t) is the number of packets sent from *s* to *t*. Multiple RBC algorithms proposed by the authors receive *G*, *R*, and *T* as input and return the RBC of all nodes. RBC is equivalent to different betweenness measures under other conditions. For example, if R(s,t,u,v) is the inverse of the number of *v*’s neighbors on the shortest path to *t* and $T\left(s,t\right)=\left\{\begin{array}{cc} 0 & s=t\\ 1 & else \end{array}\right.$, then RBC is equal to Newman’s load centrality. We assume the above T(s,t) throughout this article.

Following the common betweenness centrality notation, Dolev et al. denoted the probability of a packet passing through *v* on the way from *s* to *t* as $\delta_{st}\left[v\right]$. Therefore, an RBC vector can be computed as a sum of the $\delta_{st}$ vectors over all s,t∈V pairs.
(1)$$RBC=\sum_{s,t\in V}\delta_{st}$$

Even though RBC can mimic multiple centrality measures, defining an appropriate routing policy is a challenging task which requires a deep understanding of the target centrality measure. Our work addresses this problem by automatically learning the routing function using deep-learning techniques.

The authors discussed the combinatorial computation of RBC similar to Brandes [30]. The running-time complexity of the most efficient RBC algorithm is O(nm) when the routing policy does not depend on the packet source. When *R* depends on all four inputs, the complexity of computing RBC increases to $O(n^{2}m)$. In Section 3.4, we present an alternative differentiable computation of RBC.

### 2.4. Sound-Recognition Convolutional Neural Networks

Sound is the most common and most studied form of a wave. Many deep convolutional neural networks have been developed in recent years to recognize speech [31], emotions [32], background sounds [33], etc. The current study relies on the M5 very deep convolutional neural network proposed by Dai et al. for recognition of environmental sounds in urban areas [33]. There are two important details about their neural network architecture that should be mentioned here. First, the filter size of their first convolutional layer is set to 80, a sufficiently large value to cover the common wavelengths of natural sounds. Second, their last layers include global pooling and softmax activation function with 10 outputs to produce a classifier with 10 target classes. In Section 3.3 and Section 3.5, we replace the last layer of M5 with a single-output fully connected layer with linear activation to estimate centrality measures and routing probabilities, respectively.

## 3. Centrality-Learning Methods

### 3.1. High-Level Overview

In this article, we investigate three pipelines for learning centrality measures, see schematic description in Figure 1. The pipelines utilize two important building blocks: graph auralization  [20] used in LCGA and LRCGA pipelines and routing betweenness centrality (RBC) [34] used in LRC and LRCGA pipelines. All pipelines are used to learn arbitrary centrality measures.

In LCGA, we first produce the auralization of nodes by simulating wave propagation through the network (see Section 3.2 and Algorithm 1). Then, deep convolutions neural network for sound recognition is trained to infer centrality measures from nodes’ waveforms (see Section 3.3 and Algorithm 2). This approach is applicable to arbitrary graphs.

The second pipeline, LRC (see Section 3.4 and Algorithms  3 and 4), is only applicable to graphs with high-quality geometric embeddings for which nodes are connected if the distance between their embeddings is lower than some constant threshold. To evaluate this pipeline, we experimented with random geometric graphs (RG) [35] without relying on geometric embeddings (e.g., [36,37]) of arbitrary graphs. To generate a random geometric graph, nodes are randomly positioned in a 2D space. Then, a pair of nodes are connected if and only if their Euclidean distance is smaller than a predefined threshold. Provided node positions in the 2D space, we learned a generic routing policy R(s,t,u,v) to fit arbitrary centrality measures. *R* is computed by a fully connected neural network which receives positions of the nodes s,t,u,v and outputs the probability of *u* to forward to *v* a message sent by *s* to *t*.

The third pipeline, LRCGA, described in Section 3.5, combines LCGA with LRC. First, arbitrary graphs are auralized. Then, a deep convolutional neural network is used to infer and compute the routing probabilities from nodes’ waveforms. The neural network architecture used in LRCGA differs from the one used in LCGA, mainly because it handles four waveforms, corresponding to s,t,u,v in parallel.

Next, we elaborate on graph auralization and each one of the centrality learning pipelines.

### 3.2. Graph Auralization

In this subsection, a waveform generation process is described. Let G=(V,E) be a simple undirected unweighted graph where *V* is a set of *n* nodes and *E* is a set of *m* edges.

Consider some quantity $s_{v,t}\in R$ possessed by every node v∈V at time *t*. Intuitively, $s_{v,t}$ can be regarded as a potential of the node. $S_{t}=\left(s_{v}:v\in V\right)$ is the vector of potentials. Nodes strive to equalize their potential by distributing energy to their neighbours. Please note that, although some physical terms are used here to describe the graph analysis, they are not intended to discuss a real physical phenomenon and lack the rigorousity expected from a physics article.

Let *A* denote the adjacency matrix of the graph *G*. Let $D=\sum_{v}A_{v}$ denote the vector of node degrees. We define a right stochastic matrix $P_{u,v}=A_{u,v}/D_{u}$ as the fraction of *u*’s energy transferred to *v*. The more neighbours a node has the less energy it can transfer to each one of them.

The amount of energy every node *u* passes to every neighbor *u* at time *t* is $\Delta S_{t,u,v}=S_{t-1,u}\cdot P_{u,v}$. In matrix form,
(2)$$\Delta S_{t}=Diag\left(S_{t-1}\right)\times P,$$
where $Diag(S_{t})$ is an n×n matrix with values of $S_{t}$ along the main diagonal. Note that $\Delta S_{t,u,v}=0$ if *u* and *v* are not neighbors. Next, we introduce momentum by retaining a portion *m* of the energy flow from the previous iteration. The energy flow with momentum (from Equation (Equation 2)) is now represented by: (3)$$\Delta S_{t}=Diag\left(S_{t-1}\right)\times P+m\cdot \Delta S_{t-1}$$

The incoming energy flow of a node *v* is $\sum_{u\in V}\Delta S_{t,u,v}$ and its outgoing energy flow is $\sum_{u\in V}\Delta S_{t,v,u}$.
(4)$$S_{t,v}=S_{t-1,v}-\sum_{u\in V}\Delta S_{t,v,u}+\sum_{u\in V}\Delta S_{t,u,v}$$
Next, we will show that the energy flow defined this way can be formulated as power iterations with momentum:(5)$$S_{t}=S_{t-1}P+mS_{t-1}-mS_{t-2}=S_{t-1}\left(P+mI\right)-mS_{t-2}$$
These power iterations are similar to iterations defined by Xu et al. [38] with two exceptions. First, instead of only retaining the *m* portion of nodes’ values from the previous iteration as defined by Xu et al., Equation (Equation 5) also removes the *m* portion of nodes’ values from iteration t−2. Second, since the largest eigenvalue of *P* is one, there is no need to normalize $S_{t}$ during the interactions. See the proof of Lemma 1 for details.

**Theorem** **1**(Slowly converging energy-preserving power iterations)**.**
*Energy flow with momentum according to Equations (Equation 3) and (Equation 4) converges in connected graphs for as long as m∈[0,1) with a rate of 1/λ where λ∈[m,1].*

**Proof.** First, we will show that Equation (Equation 4) can be reformulated as power iterations with momentum. Then we will show that these iterations stabilize and $S_{t}$ is the eigenvector of *P* corresponding to the largest eigenvalue 1.Let **1** be a row vector of ones. Summing up matrix rows or columns equals multiplying the matrix by **1** on the left or by $1^{T}$ on the right. According to Equation (Equation 3):
$$\sum_{u\in V}\Delta S_{t,v,u}=\left(Diag\left(S_{t-1}\right)P+m\Delta S_{t-1}\right)1^{T}$$
$$\sum_{u\in V}\Delta S_{t,u,v}=1\left(Diag\left(S_{t-1}\right)P+m\Delta S_{t-1}\right)$$
Expanding Equation (Equation 4), we obtain:
$$S_{t}=S_{t-1}-Diag\left(S_{t-1}\right)P1^{T}+1Diag\left(S_{t-1}\right)P+1m\Delta S_{t-1}-m\Delta S_{t-1}1^{T}$$
Since $X=Diag\left(X\right)P1^{T}$ for any row vector X, $S_{t-1}-Diag\left(S_{t-1}\right)P1^{T}=0$:
(6)$$S_{t}=1Diag\left(S_{t-1}\right)P+1m\Delta S_{t-1}-m\Delta S_{t-1}1^{T}$$
Expanding $\Delta S_{t-1}$ according to Equation (Equation 3), we obtain:
$$\begin{array}{ccc} S_{t} & = & S_{t-1}P+1m\left(Diag\left(S_{t-2}\right)P+m\Delta S_{t-2}\right)-m\left(Diag\left(S_{t-2}\right)P+m\Delta S_{t-2}\right)1^{T}\\ & = & S_{t-1}P-mS_{t-2}+m\left(1Diag\left(S_{t-2}\right)P+1m\Delta S_{t-2}-m\Delta S_{t-2}1^{T}\right) \end{array}$$
Finally, we use Equation (Equation 6) to reduce the above expression and obtain Equation (Equation 5).
$$S_{t}=S_{t-1}P+mS_{t-1}-mS_{t-2}=S_{t-1}\left(P+mI\right)-mS_{t-2}$$
Note that *P* is a right stochastic matrix [39]. Rearranging the equation, we obtain a second-order matrix difference:
$$S_{t}=S_{t-1}\left(P+mI\right)-mS_{t-2}$$
In a matrix form:
$$\left[\begin{array}{c} S_{t},S_{t-1} \end{array}\right]=\left[\begin{array}{c} S_{t-1},S_{t-2} \end{array}\right]\left[\begin{array}{cc} P+mI & I\\ -mI & 0 \end{array}\right]$$
Let $A=\left[\begin{array}{cc} P+mI & I\\ -mI & 0 \end{array}\right]$. Let *λ* denote the eigenvalue of *A*. The corresponding eigenvector is $\left[\begin{array}{c} \vec{d},\vec{b} \end{array}\right]$. Therefore,
$$\left[\begin{array}{c} \vec{d},\vec{b} \end{array}\right]\left[\begin{array}{cc} P+mI & I\\ -mI & 0 \end{array}\right]=\lambda \left[\begin{array}{c} \vec{d},\vec{b} \end{array}\right],i\in \left[1,2n\right]$$
As a system of equations:
$$\left\{\begin{array}{c} \vec{d}\left(P+mI\right)-m\vec{b}=\lambda \vec{d}\\ \vec{d}=\lambda \vec{b} \end{array}\right.$$
Substituting the second equation into the first and simplifying, we obtain:
(7)$$\vec{b}P=\left(\lambda +\frac{m}{\lambda }-m\right)\vec{b}$$Let *β* denote an eigenvalue of *P*. Since we assume strongly connected graphs, according to the Perron–Frobenius theorem, *P* is irreducible. Hence, the largest eigenvalue of *P* is 1 with a unique eigenvector $\vec{\pi }$. There are two eigenvalues of *A* that satisfy $\lambda +\frac{m}{\lambda }-m=\beta$. For β=1, these are $\lambda_{1}=1$ and $\lambda_{2n}=m$. For any other β<1, λ∈(m,1). Thus, we conclude that the λ∈[m,1] and $\lambda_{1}=1$ is the largest eigenvalue of *A*.As $\vec{\pi }$ is the only eigenvector of *P* corresponding to its largest eigenvalue β=1, their multiplicity is one. There is no zero component in starting vector $\left[\begin{array}{c} S_{1},S_{0} \end{array}\right]$. Therefore, according to the power iteration method, $\left[\begin{array}{c} \vec{\pi },\vec{\pi } \end{array}\right]=\left[\begin{array}{c} \vec{\pi },\vec{\pi } \end{array}\right]A$ and $A^{t}$ converges as *t* becomes sufficiently large [40]. The convergence rate of $A^{t}$ is determined by $\frac{\lambda_{1}}{\lambda_{2}}$.If $m\to 1^{-}$, the convergence of the power iterations will be slow since $\lambda_{2}\in \left(m,1\right)$.If m>1 then it becomes the largest eigenvalue of *A* ( $m\left[\begin{array}{c} \vec{\pi },\vec{\pi } \end{array}\right]=\left[\begin{array}{c} \vec{\pi },\vec{\pi } \end{array}\right]A$). In this case, the iterations $A^{t}$ do not converge.If m=1 the iterations $A^{t}$ do not converge since $\lambda_{1}=1$ is a multiple root. □

By analyzing the eigenvalues and eigenvectors of the second-order matrix difference representation of the diffusion equation in Theorem 1, we have shown that the diffusion process converges to the stationary distribution when m∈[0,1).

The stable fixed point of the energy exchange iterations is not of interest to the current article. Let us take a close look at the dynamics of the energy exchange before the process stabilizes (see Figure 2a). The plot shows the potential levels of nodes from the graph in the first pipeline of Figure 1. On the first iteration, the potential of $v_{1}$ increases the most because it has multiple low-degree neighbours.

The potential of other nodes decreases after the first iteration because they contribute more energy than they receive. It is hard to see from this plot but the energy from node $v_{4}$ reaches $v_{1}$ and $v_{2}$ after the second iteration and then bounces back to $v_{3}$ because it is the only neighbour of $v_{4}$. Nevertheless, it is clear that the location of the nodes within the graph affects the magnitude and direction of the oscillations.

Figure 2b,c shows the oscillations with m=0.9 and m=0.99, respectively. We can see that the stabilization process is significantly prolonged. We can also see in Figure 2b irregularities caused by interference and reflection as explained in the spectral analysis literature [41].

Algorithm 1 presents the pseudo-code of graph auralization adapted for PyTorch implementation (the full source code is available on GitHub: https://github.com/puzis/centrality-learning (accessed on 23 March 2023)). The operator *T* is matrix-transpose. The operators sum and mean aggregate elements of a matrix along the dimension specified by dim. The running-time complexity of Algorithm 1 is $O(ln^{2})$ where *l* is the number of audio samples.
**Algorithm 1:** Graph auralization
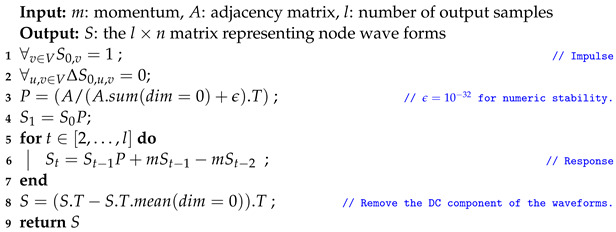


    The DC component of the output waveforms (values on which *S* stabilizes after impulse response) is not of interest. It may also hinder the convergence of sound-recognition models. Furthermore, it is a centrality measure on its own, equal to the eigenvector centrality of the graph when edges are weighted according to the inverse of the source node’s degree (considering *P* as the adjacency matrix). Thus, we removed the DC component in Line 8 of Algorithm 1 in order to show that the waveform itself bears significant information about the location of a node within the graph.

### 3.3. Directly Learning Centrality from Graph Auralization (LCGA)

#### 3.3.1. Neural Network Architecture

In the present study, the M5 classifier is adapted into a regression architecture by substituting the soft-max function with a fully connected layer. Initial experiments were conducted to evaluate the performance of different activation functions, revealing that linear activation yielded the most favourable outcomes in terms of the Pearson correlation coefficient. Other sound-recognition architectures that replace the soft-max function with a fully connected layer, a common adjustment for converting classifiers into regression models [42], were not investigated in this study. The M5 regressor is regarded as a function that maps the waveform of a node to a real-valued number, denoted as $M5:R^{l}\to R$. Consequently, the learned centrality measure of a node v∈V is represented as $M5(S_{•,v})$, where • encompasses all possible values. Furthermore, M5(S) denotes the centrality vector encompassing all nodes in the graph *G*. The structure of this algorithm is in the first pipeline of Figure 1.

The time complexity of computing the centrality of all nodes in a graph given the nodes’ waveforms is $O(ops_{M5}\cdot n\cdot l)$, where $ops_{M5}$ is the number of operations performed by the M5 neural network. Similar to the NCA-GE analysis [13], we consider $ops_{M5}$ as constant. The total complexity including the auralization is, therefore, $O\left(ln^{2}+ln\right)=O\left(ln^{2}\right)$.

#### 3.3.2. Objective Function

Let *C* denote the target centrality measure and *P* denote the predicted centrality values. The target variable for training the M5 regressor was chosen to be the Pearson correlation coefficient:$$\rho \left(C,P\right)=\frac{cov(C,P)}{std(C)\cdot std(P)}$$
The loss for training the M5 neural network is:(8)loss=1−ρ(C,M5(S))

Correlations are common performance indicators for centrality learning [12,13,15,21]. Pearson correlation coefficient has the most accessible differentiable implementation in the PyTorch deep-learning library. Other correlation coefficients can be used as long as they have differentiable implementations. A notable one is the differentiable implementation of the Spearman correlation coefficient [43].

#### 3.3.3. Training Procedure

Unlike in most deep-learning tasks, the training set in the centrality learning task is practically infinite. Random graph generation models available today can generate new data before each optimization step. However, LCGA models fail to converge when consecutive batches contain different sets of graphs. We observed this behavior with arbitrary learning rates. Models do converge quickly when they are optimized for the same small set of graphs. However, over-fitting caused by excessive optimization for the same batch makes it harder to fit a model to the next batch. Thus, the proposed training procedure (see Algorithm 2) performs several optimization steps on each batch of random graphs. During a preliminary study, we empirically identified two guidelines that increase the effectiveness of LCGA model training on random graphs:1.To mitigate the LCGA failure to converge when consecutive batches contain different sets of graphs, we performed multiple optimization steps on each batch of random graphs (lines 7–10).2.To avoid overfitting a small set of graphs, the number of graphs in a single batch should grow as the model converges (line 6).

As the model training progressed, we increased the number of random graphs in every batch in order to produce the most generic models that work on arbitrary random graphs. The constants in Algorithm 2 were chosen empirically and are subject to experiments in future research.
**Algorithm 2:** LCGA training procedure
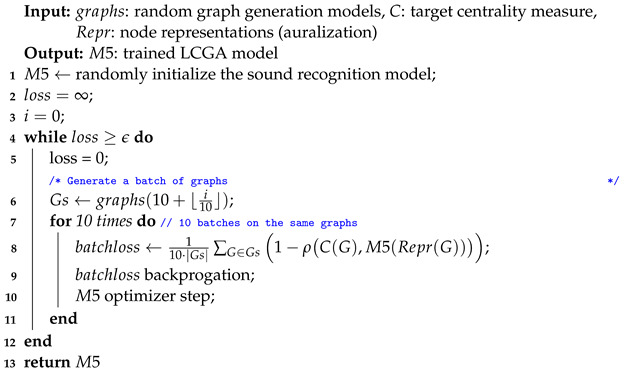


### 3.4. Learned Routing Centrality (LRC)

In this section, we present the main components of LRC relying on the geometric graph embedding as well as discussing the eigenvector RBC algorithm used to compute centrality from the routing function.

**Example** **1**(Closeness as a special case of RBC)**.**
*To showcase the feasibility of finding an appropriate routing policy to fit a given centrality measure, we provide a brief example of computing closeness using RBC with a tailor-made routing policy. Let d(s,t) be the hop distance from s to t and $CC\left(v\right)=\sum_{s}d\left(s,v\right)$ be the closeness centrality. Let $R_{CC}\left(s,t,u,v\right)$ be a routing policy that inflates the flow originating from s by one for every hop in every direction away from s: $R_{CC}\left(s,t,u,v\right)=\frac{d(s,u)+1}{d(s,u)}$. Note that $R_{CC}\left(s,t,u,v\right)$ is not a probability in this case. $\delta_{st}\left(v\right)$ is equal to d(s,t). Following Equation (Equation 1), $RBC=\sum_{s,t}\delta_{st}=\sum_{s,t}d\left(s,t\right)=n\cdot CC$.*

#### 3.4.1. The Eigenvector RBC Algorithm

Combinatorial computation of RBC is based on a topological sorting of the nodes according to the directed acyclic graph defined by R(⊗,t,u,v). Here and in the rest of this article, we use ⊗ to indicate any possible value of an argument. Topological sorting is not differentiable and, thus, cannot be efficiently utilized within deep-learning architectures to learn a routing function. In this section, we present a differentiable algebraic computation of RBC.

Consider a source node *s* and a target node *t*. R(s,t,⊗,⊗) (or R(s,t) for short) defines a stochastic Markov transition matrix leading from *s* to *t*. Following Dolev et al. [26], we require that R(s,t,s,s)=1 for all s,t∈V, forming a self-loop in every source node *s*. We also require *t* to be the only sink in R(s,t) without sink loops. We ensure that R(⊗,⊗,u,v)=0 for all u,v∉E with a scalar multiplication of R(s,t) and the adjacency matrix of the graph (*A*). The principal eigenvector of R(s,t) represents the steady-state probabilities of reaching each node on the way from *s* to *t* [44]. Thus, $\delta_{st}$ is the principal eigenvector of R(s,t), and RBC vector can be computed using Equation (Equation 1).

The algorithm for computing RBC using eigenvectors is summarized in Algorithm 3. It receives *R*, *T*, and *G* as inputs and returns the RBC of all nodes. We use the eigenvalue decomposition algorithm from PyTorch as a differentiable implementation of the $Eigenvector(R^{\prime }\left(s,t\right))$ in line 8 and assume that its complexity is $O(n^{2.376})$, as do the divide-and-conquer eigenvalue algorithms [45]. As a result, the complexity of the Eigenvector RBC algorithm is $O(n^{4.376})$. All computations described in this article were implemented using PyTorch, allowing parallelization using GPUs.
**Algorithm 3:** Eigenvector RBC
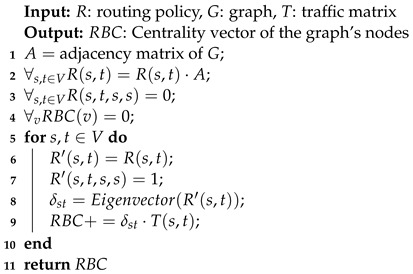


#### 3.4.2. Neural Network Architecture

Geometric graph embedding is a technique which positions nodes of a graph in a multi-dimensional Euclidean space such that nodes are connected if and only if they are at most at distance *t* for some threshold *t* [36]. Not all graphs can be embedded in a low-dimensional space. LRC pipeline is applicable only for graphs that can be. Simple heuristics, such as compass routing [46], rely on node positions to efficiently navigate the graph. When different graphs are embedded in a multi-dimensional Euclidean space, the relative positions of their nodes should follow the same geometric rules. Note that the actual positions of the nodes may be subject to arbitrary rotation and scaling. Inspired by compass routing, we developed a deep-learning-based routing function which can estimate the routing policy from node embeddings.

To simplify the notation, we denote routing policies and routing functions by *R* and use the complete terms in ambiguous contexts. Instead of explicitly fitting the 4D (i.e., s,t,u,v) routing policy to each graph, we suggest using a fully connected deep neural network as a generic routing function applicable to different graphs. This routing function estimates the routing policy from node embeddings. See the LRC pipeline in Figure 1 (middle).

The neural network receives concatenated positions of four nodes. In this article, we use 2D positions (denoted as $P^{4\times 2}$), so the input of a routing function is a vector with eight entries. In the case of geometric embedding of higher dimensions, the input length should be increased, respectively. The first two full connected layers expand the inputs and two consecutive layers reduce the dimensionality. The output is produced by a single neuron with a linear activation function and other layers use the ReLU activation function. See the neural network architecture in Figure 3. Within this figure, Dense(16,128) represents a full connected layer with 8 input features and 128 output features. *R* should be computed for all quadruples of nodes. However, the vast majority of its entries correspond to non-edges and are nullified. We compute the routing probabilities only for (s,t,u,v) entries where (u,v)∈E and use a sparse matrix to store the explicit routing policy used in Algorithm 3.

Note that unlike Dolev et al. [34], the routing policy here is not a probability. See Example 1 for a case where the routing policy $R_{CC}$ is higher than 1.

The loss function used to train the LRC depends on the task at hand. We suggest using one minus Pearson correlation as the LRC loss function, similar to Equation (Equation 8).
(9)loss=1−ρ(C,RBC(LRCNN(P),G,T)

#### 3.4.3. Training Procedure

The training procedure of LRC can be found in Algorithm 4. It is similar to the LCGA training procedure (Algorithm 2) in the sense that random graphs are constantly generated during the training process. While optimizing the routing function, there was no need for repeated optimization steps on the same graphs. Instead, we replace half of the graphs in a batch to produce the next one. We observed that such a procedure results in an acceptable trade-off between generalization and convergence speed. A rigorous investigation of the training schemes for centrality learning is out of the scope of the current article and is a subject for future research.
**Algorithm 4:** LRC and LRCGA training procedure
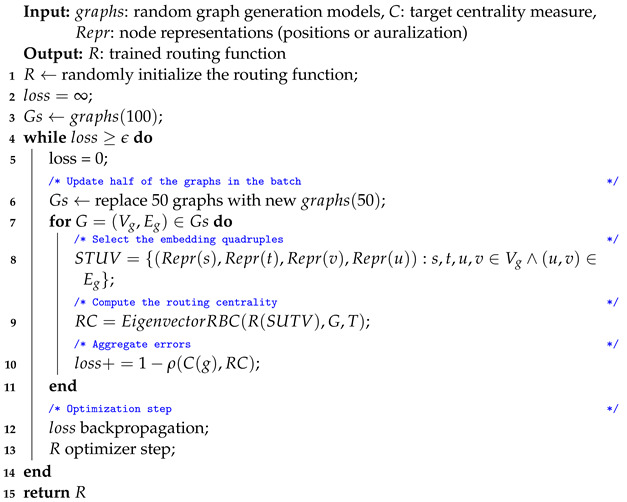


The major difference between the training procedures of LCGA and LRC is the node quadruples (line 8) and explicit computation of RBC (line 9). STUV is a collection of quadruples where each entry is a *k*-dimensional node representation. LRC is only applicable to geometric graphs; thus, node positions in Euclidean space are assumed. However, LRCGA, described next, uses the same training procedure with waveform node representations. We used the LRCNN model as the routing function (*R*) to compute the routing policy. We used Algorithm 3 to compute Eigenvector RBC from the routing policy (line 9).

### 3.5. Learning Routing Centrality Based on Graph Auralization (LRCGA)

LRC allows learning of a routing function based on node positions in 2D space because node positions and distances between nodes contain clues about routes between the nodes. We hypothesize that similar clues can be extracted from nodes’ waveforms produced by Algorithm 1. If true, this hypothesis will allow expansion of the applicability of LRC to arbitrary graph topologies. To verify this hypothesis, we combined the approaches presented in Section 3.3 (LCGA) and Section 3.4 (LRC) to learn a routing centrality based on graph auralization (LRCGA).

The bottom pipeline in Figure 1 illustrates this process. Following is the neural network architecture used to learn a routing function from nodes’ waveforms in LRCGA.

#### 3.5.1. Neural Network Architecture

Similar to the neural architecture of the LRC routing function in Section 3.4, we use here quadruples of nodes as the input. Instead of 2D coordinates, we used the first 100 samples generated by Algorithm 1 as the nodes’ representations. The quantity 100 was chosen arbitrarily to provide good-enough computation time vs. correlation trade off. The neural network comprises two parts: (1) a 1D convolutional network and (2) a fully connected network.

See Figure 4 for the complete architecture. Black numbers in the figure indicate the size of the data. Red marks present the layers’ configurations. For example, Conv1d(3,2) defines a 1D convolution layer with a kernel size of 3 and stride of 2. MaxPool1d(2,2) denotes a 1D max pool layer with a kernel size of 2 and stride of 2. Dense(16,128) means a full connected layer with 8 input features and 128 output features. Note that, except for the output, we used the ReLU activation function in all layers. The output layer uses a linear activation.

The 100-sample waveforms of the four input nodes correspond to four channels of the 1D convolution. The first part of the network generates 16 features by reducing the dimensionality of the 4×100 input through three pairs of Conv1D and MaxPool layers. The second part comprises fully connected layers similar to the routing function for LRC in Section 3.4. The fully connected network learns the intrinsic relations between (s,t,u,v), outputting the routing function value.

#### 3.5.2. The LRCGA Pipeline

Similar to LRC, we began the training process by recording the waveforms of all reasonable combinations of (s,t,u,v). Every tuple (s,t,u,v) of auralized nodes is transformed by the neural network into a routing policy value and stored in a sparse graph. Finally, we used Algorithm 3 to obtain the centrality vector of the whole graph. Similar to LRC, since the routing function receives node representations as an input, the entire process can be applied to arbitrary graphs of various sizes.

The training procedure is similar to the LRC pipeline except that LRCGA is trained with multiple random graph models. During the training process, the output centrality vector is compared to the ground truth and the parameters of the routing function are updated in each optimization step. We use Pearson correlation coefficient to compute the difference between ground-truth centrality and output centrality:(10)loss=1−ρ(C,RBC(LRCGANN(S),G,T)

The time complexity of computing the centrality of all nodes in a graph using the LRCGA includes auralization $O(ln^{2})$, routing-policy inference $O(n^{2}m\cdot ops_{LRCGANN}$), and EigenvectorRBC computation $O(n^{4.376})$. The total running-time complexity of LRCGA on a single graph is $O(ln^{2}+n^{4.376})$.

## 4. Empirical Evaluation and Demonstration

Next, we evaluated the centrality-learning pipelines presented in Section 3. The general experimental setup is described in Section 4.1. Then, we present benchmark results in Section 4.2 comparing the proposed methods with each other and with the state-of-the-art NCA-GE [13]. In Section 4.3, we dive deeply into the performance of LCGA.

### 4.1. Experimental Setup

In this subsection, we consider the general task of centrality learning. Connectivity degree (Deg), closeness centrality (CC), eigenvector centrality (EC), load centrality (LC) and betweenness centrality (CC) are considered as the target centrality measures to demonstrate the learning process.

It is evident from past research [12,13,14] and from the results in this article that certain kinds of graphs are harder to learn than others. Our findings indicate that, overall, the BA and ER random graph models outperform other models, owing to variations in the topology of different random graph models. Therefore, LCGA and LRCGA were trained on four different random graph generation models: the Erdos–Renyi (ER) random graph, Barabasi–Albert (BA) scale-free graph, Watts–Strogatz (WS) small-world graph, and random geometric graphs (RG) shown in Figure 5a–d. LRC was only evaluated on RG due to its design limitation.

#### 4.1.1. Test Data

New random graphs are generated during the testing phase. Three types of test sets are considered:Small random graphs—similar to the graphs generated during the training process.Larger random graphs—generated using the same models but containing more nodes.Real graphs—four famous graphs as depicted in Figure 5e–h.

LRC was evaluated using the RG models only. Models trained on the small graphs were applied to the large graphs as well. For LCGA and LRCGA, this is possible due to the graph-size-invariant length of the nodes’ representation using waveforms (e.g., l=100). Note that the waveforms of larger graphs usually exhibit more frequencies (a richer sound) corresponding to a larger number of eigenvalues. For LRC, the node representation length depends on the dimensionality of the geometric embedding and is size-invariant as well.

#### 4.1.2. Performance Indicators

We computed the Pearson, Spearman, and Kendal correlation coefficients between the ground truth and learned centrality measures. Since the training procedures optimize for Pearson correlation, it is expected to exhibit the best results.

### 4.2. Benchmark Results

Three methods proposed in this article are compared with each other and with the state-of-the-art NCA-GE [13] as a baseline. Benchmark results are presented in Table 1. The best performance is marked with boldface. The correlation with degree centrality is almost perfect due to the simplicity of this centrality measure. LCGA and LRCGA perform preferably in all cases, indicating the benefits of the waveform representation. LRC results are presented only for the RG graphs due to the design limitation of LRC. NCA-GE was not compared with the degree centrality because it uses degree as a feature. All models show the worst performance on the WS networks. We attribute this behavior to the high regularity of WS networks leading to the structural similarity of many nodes.

Table 2 presents the benchmark results on four real graphs. In these graphs, the performance of LCGA and LRCGA is consistently better than the performance of NCA-GE. All models perform the worst on the Les-Miserables graph. We found no appealing explanation for this behaviour. Explainable artificial-intelligence (XAI) methods that are applicable to GNN [54] could help identify the topological caveats leading to such misperformance. Yet, such XAI methods need to be adapted for the auralization pipeline.

### 4.3. Auralization Showcase and LCGA Performance

#### 4.3.1. The Voice of Nodes

The echo chambers formed by graphs produce sound patterns, as can be seen and heard in the multimedia Figure 6. On the left side is the spectrum of nodes. While peak frequencies remain the same for different nodes in the graph, their intensities vary in line with spectral graphs theory. The centrality-learning approach of LCGA and LRCGA relies on these differences.

#### 4.3.2. The Learning Process

Next, we demonstrate the training procedure of LCGA. Figure 7 shows the training loss with learning nodes’ connectivity degree from their auralization. It is clear that node auralization contains information about connectivity degree. The loss decreases consistently with the number of optimization steps. In Figure 7 (left), we can observe a monotonic decrease in loss during 10 optimization steps on each batch. Every new batch results in a sharp decrease in the loss. However, on average, the loss gradually decreases, indicating an efficient learning process.

#### 4.3.3. Scalability of LCGA

Training deep neural networks requires significant computational effort. Thus, a deep-learning-based centrality-learning method should be scalable to facilitate applications on graphs larger than those on which it was trained. We finish the empirical evaluation in this article by showing that LCGA, trained on small graphs with 150 nodes, performs equally well on larger graphs with 1500 nodes. Table 3 presents the Pearson correlation coefficients of LCGA on large graphs. Naturally, the degree is the easiest to learn. The correlation coefficients for other centrality measures are also as high as on the small graphs LCGA was trained with, except WS. The limitations of LCGA are highlighted using WS graphs. Not only do these graphs result in the worst performance across all the centrality-learning models evaluated in this study, we also see the highest degradation in performance when testing LCGA with large WS graphs. Yet the correlation values produced by LCGA for WS graphs across all centrality measures are generally higher than correlations between the centrality measures [21]. Since centrality measures are inherently correlated, a good learned centrality measure should correlate better with the one it has learned from than with other centrality measures. This fact suggests that the waveforms produced by Algorithm 1 effectively encode information about the position of nodes within the graphs.

## 5. Conclusions

This article presents three methods to learn centrality—LCGA, LRC and LRCGA. Among them, LCGA and LRCGA are suitable for all kinds of graphs. LRC only works on random geometric graphs. LCGA performs impressively well on graphs larger by an order of magnitude than the graphs on which it was trained. This article also demonstrates by an example that the waveforms produced by graph auralization contain information about nodes’ positions within the graph and can be used for inference and classification.

There is still a lot of work to do to improve the proposed pipelines. First, it is important to reduce the computational resources for the training of LRCGA. This challenge may be achieved by sampling only some s,t,u,v quadruples during the training process. Second, the reported results indicate that centrality learning is harder on some kinds of graphs than on others. Future research should investigate the challenges of centrality learning in models such as WS small-world networks and develop better, generic learning approaches. A solution to both challenges may come from the development of flexible embedding methods which learn tailor-made node representations. Future work includes investigating a transfer learning problem where centrality learned from one kind of graph (e.g., ER) is adjusted to fit graphs with significantly different structures (e.g., WS). Future work can explore the discovered graph auralization phenomenon as a means to learn node properties. The study of auralization methods and the development of the respective theory is an area that warrants further research. This paper also highlights that the number and the mix of graphs within mini-batches are important for efficient training. Further investigation in this direction can produce efficient distillation strategies that optimize the set of training graphs. Future work also involves the realization and evaluation of various loss functions, such as Mutual Information and Kendall, for centrality learning. A library of differentiable correlation functions will contribute not only to learning accurate centrality measures but to a variety of deep-learning regression tasks.

## Figures and Tables

**Figure 1 entropy-25-01115-f001:**
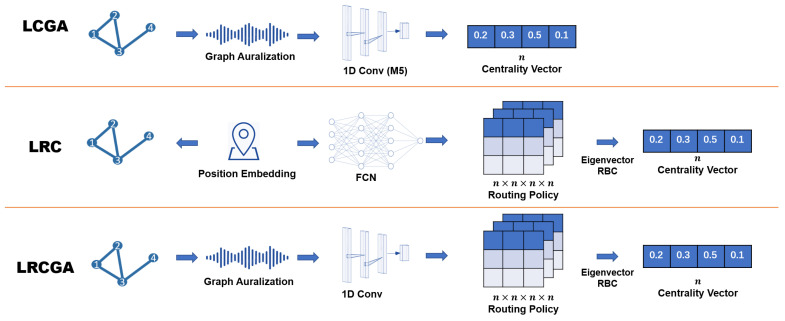
Centrality learning pipelines. LCGA: graph auralization and direct centrality inference nodes’ waveforms. LRC: routing policy is inferred from node positions to fit a given centrality measure. LRCGA: graph auralization and routing policy are inferred from nodes’ waveforms to fit a given centrality measure.

**Figure 2 entropy-25-01115-f002:**
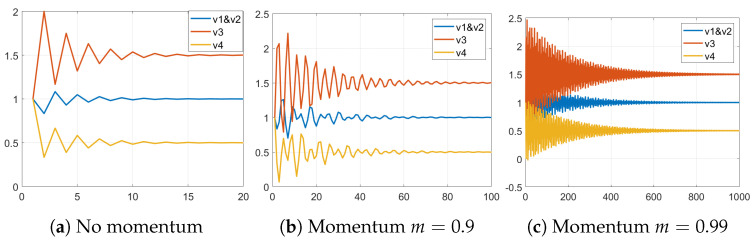
Energy exchange interactions with various levels of momentum.

**Figure 3 entropy-25-01115-f003:**
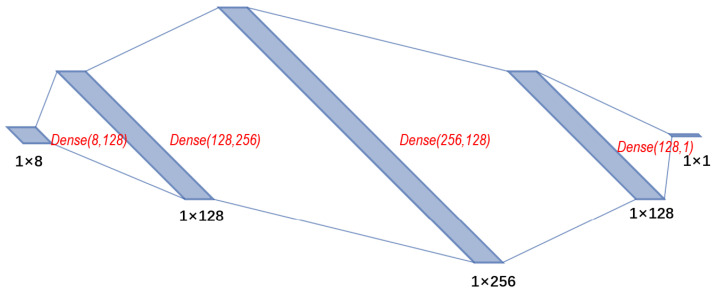
LRCNN neural network architecture of the routing function for LRC.

**Figure 4 entropy-25-01115-f004:**
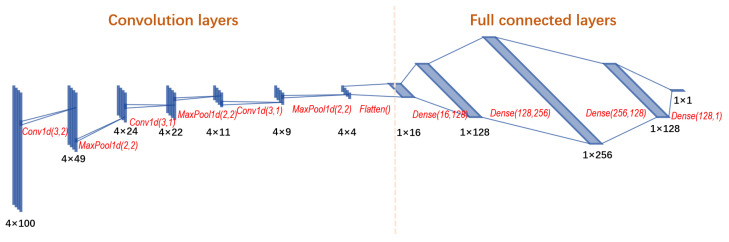
LRCGANN: neural network architecture of the routing function for LRCGA.

**Figure 5 entropy-25-01115-f005:**
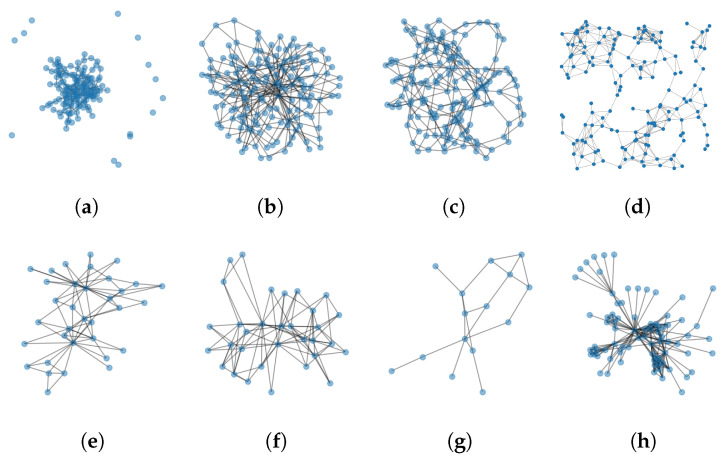
Examples of some random and well-known graphs. On the top (**a**–**d**): Erdos–Renyi (ER) random graph [47], Barabasi–Albert (BA) scale-free graph [48], Watts–Strogatz (WS) small-world graph [49], random geometric graphs (RG) [35]. On the bottom (**e**–**h**): Karate club [50], Southern women graph [51], Florentine families graph [52], and Les Miserables [53].

**Figure 6 entropy-25-01115-f006:**
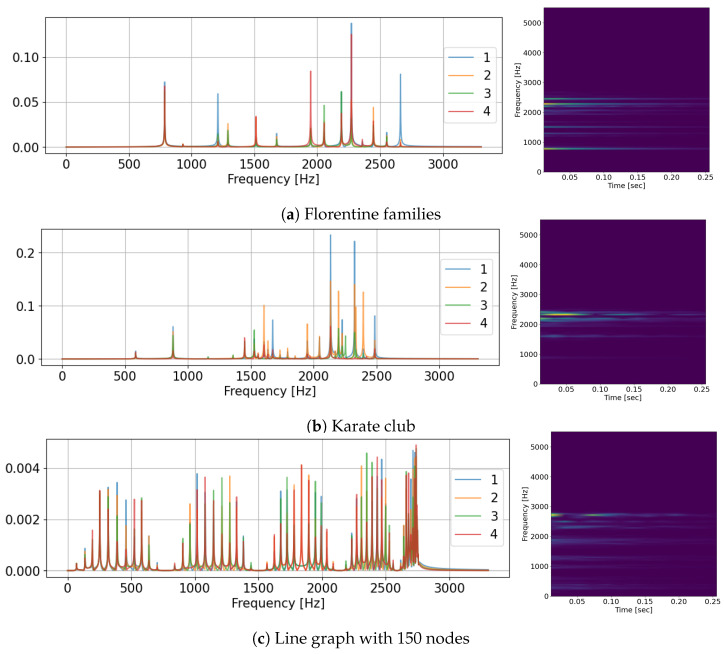
The voice of graphs and nodes. **Left**: the spectra of four nodes in a graph. **Right**: a spectrogram of one arbitrary node.

**Figure 7 entropy-25-01115-f007:**
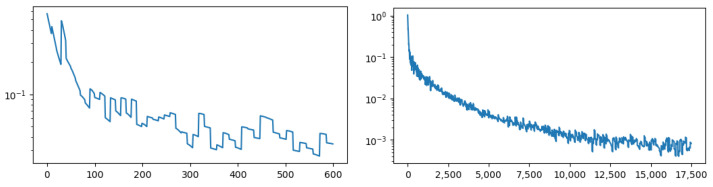
The loss (1−ρ(M5(S))) as a function of the number of optimization steps when learning the connectivity degree. **Left**: 50 batches. **Right**: 500 batches.

**Table 1 entropy-25-01115-t001:** Centrality learning benchmark on random graphs with 150 nodes each. Maximal correlation scores are highlighted in bold.

		Betweenness	Closeness	Load	Degree	Eigenvector
RG	NCA-GE	0.58	0.91	0.62	/	0.89
	LCGA	0.92	**0.97**	**0.95**	**0.99**	**0.99**
	LRC	0.85	0.93	0.88	**0.99**	**0.99**
	LRCGA	**0.93**	0.94	0.82	**0.99**	0.98
BA	NCA-GE	0.96	0.96	0.96	/	0.96
	LCGA	**0.99**	0.89	0.95	**0.99**	0.96
	LRC	/	/	/	/	/
	LRCGA	0.95	**0.97**	**0.98**	**0.99**	**0.99**
WS	NCA-GE	0.89	0.85	0.89	/	0.87
	LCGA	0.84	0.82	**0.97**	**0.99**	0.77
	LRC	/	/	/	/	/
	LRCGA	**0.93**	**0.94**	0.94	**0.99**	**0.99**
ER	NCA-GE	**0.98**	**0.97**	0.98	/	0.97
	LCGA	0.95	**0.97**	**0.99**	**0.99**	0.93
	LRC	/	/	/	/	/
	LRCGA	0.92	0.96	0.94	**0.99**	**0.99**

**Table 2 entropy-25-01115-t002:** Centrality-learning benchmark on real graphs. Maximal correlation scores are highlighted in bold.

		Betweenness	Closeness	Load	Degree	Eigenvector
Karate club	NCA-GE	0.71	0.68	0.73	/	0.67
	LCGA	**0.94**	**0.89**	**0.97**	**0.99**	0.91
	LRCGA	0.90	0.82	0.92	**0.99**	**0.99**
Southern women	NCA-GE	0.88	0.77	0.88	/	0.93
	LCGA	**0.98**	0.71	**0.99**	**0.99**	0.88
	LRCGA	**0.98**	**0.91**	0.97	**0.99**	**0.99**
Florentine families	NCA-GE	0.72	0.71	0.73	/	0.90
	LCGA	0.93	0.68	**0.92**	**0.99**	0.90
	LRCGA	**0.95**	**0.82**	0.89	**0.99**	**0.99**
Les Miserables	NCA-GE	0.70	0.67	0.70	/	0.93
	LCGA	0.75	0.39	**0.98**	0.87	0.75
	LRCGA	**0.90**	**0.75**	0.84	**0.96**	**0.98**

**Table 3 entropy-25-01115-t003:** Performance of the LCGA pipeline trained on graphs with 150 nodes and tested with 1500 nodes.

	Betweenness	Closeness	Load	Degree	Eigenvector
RG	0.97	0.97	0.95	0.99	0.99
BA	0.97	0.92	0.88	0.99	0.96
WS	0.70	0.65	0.97	0.99	0.69
ER	0.98	0.77	0.99	0.99	0.83

## Data Availability

The source code of the algorithms and experiments reported in this article is available on GitHub https://github.com/puzis/centrality-learning.git (accessed on 23 March 2023).

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
