# Peer review of "Centrality Learning: Auralization and Route Fittingâ€"

_entropy, 2023, doi:10.3390/e25081115_

Round 1
Reviewer 1 Report
The authors learn deep neural networks to asses graph centrality measures. Advantages of the article: 1) A good literature review. 2) Proposition of the deep neural network. 3) Implementation of the network, training and learning. Disadvantages of the article: 1) The reader needs help to recreate the results obtained by the authors easily. The code is not obtainable on the Internet.
Author Response
Dear Editors, Reviewers,
Entropy, MDPI
Thank you for your valuable and constructive comments that led to significant improvement of the article. We addressed most of the comments and suggestions in the revised manuscript. The changes we incorporated and responses to the comments are listed next. The changes in the manuscript text are marked with blue in this response letter and the submitted revised manuscript.
Best regards,
The authors'
Reviewer 1:
Comment 1: The reader needs help to recreate the results obtained by the authors easily. The code is not obtainable on the Internet.
Reply: The URL in the code/data availability statement (https://github.com/puzis/centrality-learning) is leading to a public GitHub repository where any user can access the code. We welcome feature requests, bug reports, and pull requests.
Reviewer 2 Report
After reviewing the article "Centrality learning: auralization and route fitting" I can make the following remarks.
The article makes a very good impression: it is well thought out, well organized, has undoubted novelty and practical significance. The article is written in good English. I did not see any flaws in the formatting of the work.
I was very interested in this work. The idea of applying auralization to the graph centrality problem is a very good one.
However, I have a few notes to work with:
1. The meaning of the graph centrality metric and why it should be measured at all is not completely clear from the article.
2. It would be good for the authors to take a deeper look at other methods used for graph embedding. For example, graph neural networks [https://doi.org/10.1109/TNN.2008.2005605], node2vec [https://doi.org/10.48550/arXiv.1607.00653], etc.
3. I was surprised by the choice of graphs for analysis. Why were random graphs and some strange graphs built on the basis of the relationships between the characters of fiction book taken?
Why did the authors not use any well-known datasets, for example, TUDataset [https://doi.org/10.48550/arXiv.2007.08663], FB15k [https://dl.acm.org/doi/10.5555/2999792.2999923], etc. [https://doi.org/10.48550/arXiv.2005.00687].
Or did they not take topology graphs from some application area, for example, communication networks?
Author Response
Dear Editors, Reviewers,
Entropy, MDPI
Thank you for your valuable and constructive comments that led to significant improvement of the article. We addressed most of the comments and suggestions in the revised manuscript. The changes we incorporated and responses to the comments are listed next. The changes in the manuscript text are marked with blue in this response letter and the submitted revised manuscript.
Best regards,
The authors'
Reviewer 2:
Comment 1: The meaning of the graph centrality metric and why it should be measured at all is not completely clear from the article.
Reply: Thank you. We now illustrate the importance and rationality of graph centrality in section 1 (introduction):
[...] Centrality computation is one of the most important and commonly used tools in the analysis of complex networks [5,6]. Centrality measures are properties of nodes that correspond to the nodes’ importance for different tasks. For example, nodes with high betweenness centrality can be used to monitor network flows [7,8], nodes with high closeness can easily reach all other network participants, they can be used to disseminate information [9], etc.
Comment 2: It would be good for the authors to take a deeper look at other methods used for graph embedding. For example GNN and node2vec.
Reply: Unfortunately our literature review did not surface papers describing successful attempts for learning arbitrary centrality measures using generic graph embedding methods such as Deepwalk, node2vec, LINE, Net-LSD, and others. In our preliminary experiments, we also were not successful with learning centrality directly from such node embeddings. Therefore, they are not deeply covered in the current paper.
We mention graph embedding techniques such as Structure2Vec used in conjunction with other methods to learn centrality. For example, the work by Mendonca et al. in section 2.1. We take a deeper look at GNN. The whole of Section 2.2 is dedicated to GNN-based approaches for centrality learning.
Comment 3: Why were random graphs and some strange graphs built on the basis of the relationships between the characters of the fiction book taken?
Reply: We use the most known random graph models Erdos-Renyi (ER) random graph, Barabasi-Albert (BA) scale-free graph, and Watts-Strogatz (WS) small-world graph to perform the experiments. We also use some well-known real-world graphs such as Karate Club, Southern Women graph, Florentine Families graph and Les Miserables graph.
Comment 4: Why did the authors not use any well-known datasets, for example, TUDataset [https://doi.org/10.48550/arXiv.2007.08663],
FB15k [https://dl.acm.org/doi/10.5555/2999792.2999923],
etc. [https://doi.org/10.48550/arXiv.2005.00687].
Or did they not take topology graphs from some application area, for example, communication networks?
Reply: Indeed there are many graph datasets some of which were rightfully mentioned by the reviewer. We use a few of the most known graphs for the evaluation. Karate club [50] (5800 citations), Southern women graph [51] (2021 citations), Florentine families graph [52] (194 citations) and Les Miserables graph [53] (1428 citations). Unfortunately, the 10 days we received for the manuscript revision are not sufficient to reproduce the experiments with additional graph datasets.
Reviewer 3 Report
This paper reports on a very interesting architecture which uses the transformation of graphs into a sound structure that is then used for the prediction of some centrality characteristics. It is a very untypical pipeline with lots of interesting facts and ideas for future work. The researchers first introduce their ideas thoroughly with more than enough background work and present the architectures of choice with some examples and waveforms generated by the graphs. The authors provide a GitHub repository with their code, analysis and comparison with similar works, present the results of their experiments and have some interpretations also.
The paper does have originality because it is quite novel to transform a graph to sound (auralization) and then deduce particular properties from it. It is quite interesting if sonification strategies could be applied to it too and helped infer other properties as well.
Some references are missing, such as in line 217 the power iterations method needs one. Turning a classifier into a regression model (lines 262-263) needs also a reference. For explanation purposes, the following methods can be used:
- Holzinger, A., Saranti, A., Molnar, C., Biecek, P., & Samek, W. (2022, April). Explainable AI methods-a brief overview. In xxAI-Beyond Explainable AI: International Workshop, Held in Conjunction with ICML 2020, July 18, 2020, Vienna, Austria, Revised and Extended Papers (pp. 13-38). Cham: Springer International Publishing.
https://doi.org/10.1007/978-3-031-04083-2_2
- - Schnake, T., Eberle, O., Lederer, J., Nakajima, S., Schütt, K. T., Müller, K. R., & Montavon, G. (2021). Higher-order explanations of graph neural networks via relevant walks. IEEE transactions on pattern analysis and machine intelligence, 44(11), 7581-7596.
https://doi.org/10.1109/tpami.2021.3115452
Overall, the methodology is sound. The reviewers only want to highlight some parts that are not clear to them. In line 117, what were the different pre-processing steps and what were the different GNN structures? What are the differences between RBC, SPBC, FBC and LC? The authors present the time complexity of other algorithms in lines 157-160; it would be profitable if they could explain what the time complexity of their solution is too. What were the assumptions about the results of the experiments as far as the referred high-quality geometric embeddings in line 186? The claim in lines 190-191 needs an example for understanding since it is not so intuitive for the reader. The sentences in lines 205-207 are highly appreciated; the same goes for the GitHub repository. A very important question on page 6 is: will it necessarily stabilize? The irregularities mentioned in line 236 need a corresponding reference. Are there tests made with the DC component? This correspondence with eigenvector centrality needs to be shown or a reference is needed. The use of Pearson correlation is highly questionable because it only detects linear correlations. Measurements were made also with Spearman and so on, but the training should not be made only with this one; Mutual Information (MI) needs also to be considered. Why are LCGA models fail to converge? The empirical choices (line 280) need explanations with Explainable AI (xAI) methods. It is not clear how the researchers came up with the ideas in lines 275-277. The whole section 3.4.2. is questionable. The non-incorporation of the 3D structure (or relative position) of the nodes is not adequate in the era of geometric deep learning. Why is it claimed that the actual positions of the nodes do not matter? Is that checked? If it doesn’t there is no need of mentioning the 3D embeddings. Without the use of xAI and corresponding experiments, this cannot be said with such certainty. Why is the loss 1 – Pearson correlation? The claims in lines 341-342 need to be shown and validated with xAI methods. Concatenating the positions of the nodes is a bad practice in this case probably, since they don’t represent a continuous variable (to be concatenated). It is irritating for the reader why they are used and then it is said that they don’t play a decisive role; this needs to be explained.
The trade-off between generalization and convergence in line 381 is missing. Was the hypothesis in line 364 verified? The use of the first 100 samples (line 373) is questionable; are the data shuffled? The use of accuracy is not suggested in general, mostly in imbalanced datasets (line 375). The use of Mutual Information is suggested. All results of lines 378-390 are not complete without xAI; the same goes with the claim of lines 413-414 – which ones are harder? The claims in lines 426-428 are not that straightforward, they need to be shown. In line 446 the authors write that all models perform worst on the Les Miserables graph without explanation, but also without the use of xAI. If they had used one or two xAI methods, they could find why. The reliance on differences in line 453 can also be shown with those methods. The difference between „all“ centrality measures and „between“ needs a better explanation in lines 474-476.
The paper is well-written, clear and readable for the most part. The reviewers did not find any typos. Some expressions need a more extended explanation, such as „not break gradient back-propagation“ (line 56).
Round 2
Reviewer 2 Report
My impression of the article remained the same:
The article makes a very good impression: it is well thought out, well organized, has undoubted novelty and practical significance. The article is written in good English. I did not see any flaws in the formatting of the work.
All my comments were taken into account and exhaustive answers were given.
I recommend this article.
Reviewer 3 Report
The changes that were made are adequate. I went through the GitHub repository and read also the thesis file. That is a nice work. Adding mathematical details and future work helped the acceptance of the paper.